# Ferredoxin 1 is essential for embryonic development and lipid homeostasis

Shakur Mohibi[1†], Yanhong Zhang[1†], Vivian Perng[1], Mingyi Chen[2], Jin Zhang[1*], Xinbin Chen[1*]

[1]Comparative Oncology Laboratory, Schools of Veterinary Medicine and Medicine, University of California, Davis, Davis, United States; [2]Department of Pathology, University of Texas Southwestern Medical Center, Dallas, Dallas, United States

**Abstract** Mammalian ferredoxin 1 and 2 (FDX1/2) belong to an evolutionary conserved family of iron-sulfur cluster containing proteins and act as electron shutters between ferredoxin reductase (FDXR) and numerous proteins involved in critical biological pathways. FDX1 is involved in biogenesis of steroids and bile acids, Vitamin A/D metabolism, and lipoylation of tricarboxylic acid (TCA) cycle enzymes. FDX1 has been extensively characterized biochemically but its role in physiology and lipid metabolism has not been explored. In this study, we generated *Fdx1*-deficient mice and showed that knockout of both alleles of the *Fdx1* gene led to embryonic lethality. We also showed that like *Fdxr*+/-+/-, *Fdx1*+/-+/- had a shorter life span and were prone to steatohepatitis. However, unlike *Fdxr*+/-+/-, *Fdx1*+/-+/- were not prone to spontaneous tumors. Additionally, we showed that FDX1 deficiency led to lipid droplet accumulation possibly via the ABCA1-SREBP1/2 pathway. Specifically, untargeted lipidomic analysis showed that FDX1 deficiency led to alterations in several classes of lipids, including cholesterol, triacylglycerides, acylcarnitines, ceramides, phospholipids and lysophospholipids. Taken together, our data indicate that FDX1 is essential for mammalian embryonic development and lipid homeostasis at both cellular and organismal levels.

*For correspondence:
jinzhang@ucdavis.edu (JZ);
xbchen@ucdavis.edu (XC)

†These authors contributed equally to this work

Competing interest: The authors declare that no competing interests exist.

## Editor's evaluation

The findings provided by Mohibi et al. are important to the field of lipid metabolism and cancer and provide insight for an in vivo role of FDX1. The evidence is solid, utilizing multiple modalities and both in vitro and in vivo lines of investigation.

## Introduction

Ferredoxin (FDX) proteins are a family of redox proteins present in all branches of life (*Schulz et al., 2023*). They contain iron-sulfur (Fe-S) clusters that aid in their function of transferring electrons from NADPH via ferredoxin reductase (FDXR) to systems involved in various biological processes including iron sulfur (Fe-S) cluster synthesis, steroidogenesis, bile acid production and vitamin metabolism. Mammalian cells contain two ferredoxins, FDX1 and FDX2, both of which are localized to mitochondria (*Sheftel et al., 2010*). Although they have high homology in primary and three-dimensional structures, they differ significantly in their substrate identity and function (*Schulz et al., 2023*). FDX2 is critical in the formation of Fe-S clusters in mitochondria by donating electrons to iron sulfur cluster assembly complex (*Schulz et al., 2023*). FDX1, on the other hand, donates its electrons to various cytochrome p450 systems present in mitochondria and is involved in production of steroid hormones from cholesterol (steroidogenesis), bile acid production, addition of heme α to Cytochrome c as well as Vitamin A/D metabolism (*Schulz et al., 2023*). Recently, FDX1 has been shown to regulate the process of lipoylation of tricarboxylic acid (TCA) cycle enzymes, especially pyruvate dehydrogenase

(PDH) and a-ketoglutarate dehydrogenase (*Schulz et al., 2023* ; *Tsvetkov et al., 2022*). Lipoylation is essential for the function of these enzymes. Moreover, FDX1 has emerged as the central player in the regulation of a new form of cell death, called cuproptosis (*Tsvetkov et al., 2022*; *Tsvetkov et al., 2019*).

Lipids consist of a myriad group of biomolecules essential for various cellular processes, including membrane formation, energy storage and cellular signaling pathways (*Snaebjornsson et al., 2020*). Some vital lipids, such as triaclyglycerides (TAGs), cholesterol, cardiolipins (CLs) and phospholipids, are essential for cell survival. Due to the essential role of lipids in maintaining normal physiology, aberrant lipid homeostasis has been involved in various disease pathologies, including cancer, neuro-degeneration, and cardiovascular diseases (*Broadfield et al., 2021*; *Hoy et al., 2021*; *Snaebjornsson et al., 2020*). Altered lipid metabolism is also linked with systemic inflammation and non-alcoholic steatohepatitis (NASH) (*Carotti et al., 2020*; *Hoy et al., 2021*; *Pei et al., 2020*; *Singla et al., 2010*).

SREBP1/2 are considered master regulators of lipid metabolism and often associated with altered lipid homeostasis (*Osborne and Espenshade, 2009*; *Shao and Espenshade, 2012*). In physiological conditions, when intracellular cholesterol levels are depleted, SREBP1/2 are acti-vated through the release of their N-terminal domains by proteolytic cleavage. The N-terminal domains are then translocated to nucleus where a set of genes involved in lipid metabolism are induced by SREBP1/2 (*Osborne and Espenshade, 2009*; *Shao and Espenshade, 2012*). However, in diseased conditions, SREBP1/2 are constitutively activated via various mechanisms, resulting in abnormal lipid production and accumulation (*Osborne and Espenshade, 2009*; *Shao and Espen-shade, 2012*). SREBP1 is expressed as two isoforms, SREBP 1 a and 1 c, both of which are tran-scribed from the SREBF1 gene but through two separate promoters. SREBP-2 is encoded by the SREBF2 gene. SREBP-1c preferentially regulates genes for synthesis of fatty acids (FAs) and TAGs whereas SREBP-2 preferentially regulates genes for cholesterol synthetic pathway, including the mevalonate pathway (*Brown and Goldstein, 1997*). SREBP-1a has overlapping functions with both SREBP-1c and SREBP-2.

FDXR transfers electrons from NADPH to FDX1, which then delivers the electrons to either COX15, lipoyl synthase or seven different mitochondrial cytochromes, thus supporting the biosynthesis of heme α, lipoic acid, steroid hormones, bile acid, and Vitamin A/D metabolism (*Schulz et al., 2023*). We previously showed an essential role of FDXR in mammalian embryonic development and lipid metabolism (*Zhang et al., 2022*; *Zhang et al., 2017*). As FDX1 is a substrate of FDXR in regulating steroidogenesis, we asked whether FDX1 is necessary for mammalian embryonic development and lipid metabolism. We showed that mice homozygous for *Fdx1* deletion are embryonically lethal at embryonic day 10.5 (E10.5). We also showed that *Fdx1*[+/-] +/- have a short life span and are prone to steatohepatitis, but not to spontaneous tumors. Like FDXR, FDX1 was found to regulate lipid metab-olism due to increased activation of the SREBP1/2 pathways. Moreover, untargeted lipidomic analysis showed that FDX1 loss leads to alterations in several classes of lipids.

## Results

### *Fdx1*[-/-] embryos die between E10.5 and E13.5

Although FDX1 has been extensively studied biochemically, its role in development and physiology has been unexplored. To determine physiological function of FDX1, we generated *Fdx1* heterozygous mice using the knock-out first approach (*Figure 1A and B*). The *Fdx1*[+/-] +/- developed normally and were fertile. However, inter-breeding of *Fdx1*[+/-] +/- to obtain *Fdx1*[-/-] mice was unsuccessful. Out of the 67 pups obtained from the *Fdx1*[+/-] +/- inter-breeding, none were of the *Fdx1*[-/-] genotype (*Figure 1C*), suggesting complete loss of *Fdx1* leads to embryonic lethality. To determine at what developmental stage the embryonic lethality occurs, we isolated embryos following timed pregnancies. We did not find any *Fdx1*[-/-] embryos at embryonic day 13.5 (E13.5) but found one at E10.5 (*Figure 1C*), suggesting *Fdx1*-KO is lethal for embryos between E10.5-E13.5. Although, the size of WT and *Fdx1*[+/-] +/- at E10.5 was comparable, *Fdx1*[-/-] embryos were small and abnormal (*Figure 1D and E*). Interestingly, this embryonic lethality is commensurate with that observed with the embryonic lethality of *Fdxr*-KO at E8.5 (*Zhang et al., 2017*). These results suggest that Fdx1 and Fdxr regulate similar processes required for mammalian embryonic development and survival.

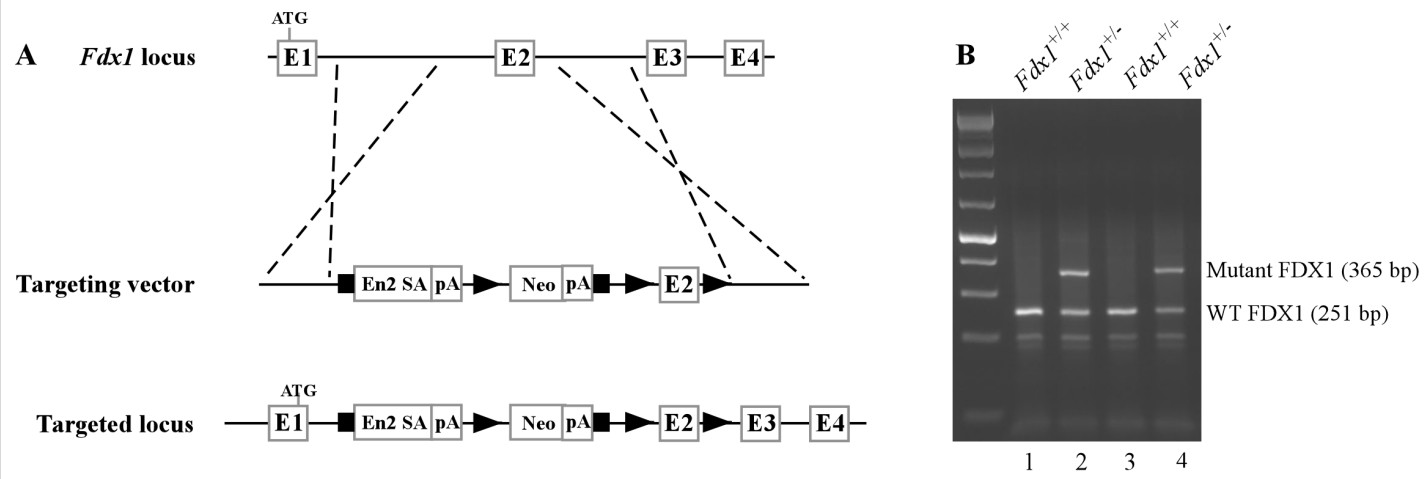

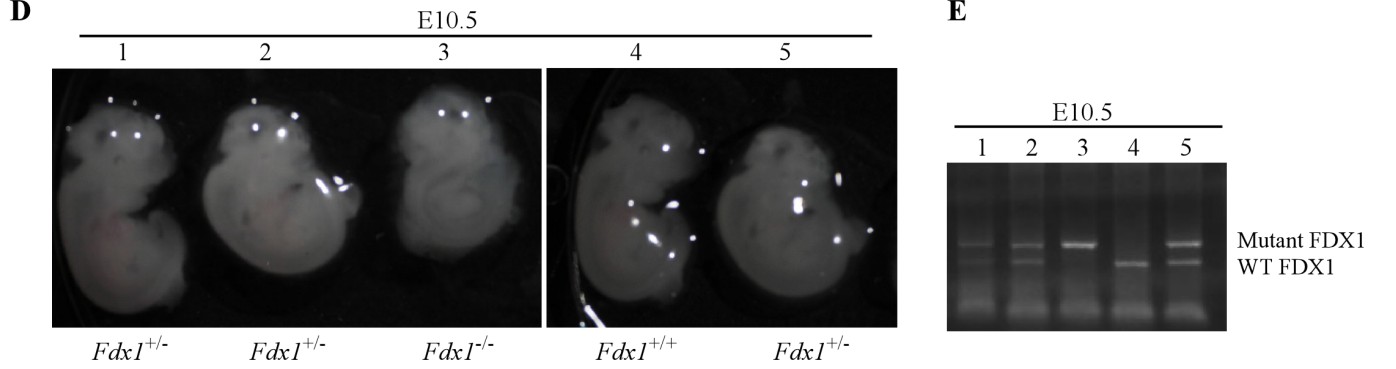

**C** The number and percentage of mice and embryos from the intercross of $Fdx1^{+/-}$ mice

| Stage | $Fdx1^{+/+}$ (25%) | $Fdx1^{+/-}$ (50%) | $Fdx1^{-/-}$ (25%) | Resorbed | Total# |
|---|---|---|---|---|---|
| Live mice | 32 (47.7%) | 35 (52.3%) | 0 (0%) | | 67 |
| E13.5 | 7 (33.3%) | 9 (56.3%) | 0 (0%) | 5 (31.3%) | 21 |
| E10.5 | 1 (14.4%) | 3 (42.9%) | 1 (14.4%) | 2 (28.9%) | 7 |

**Figure 1.** $Fdx1$ knock-out mouse are embryonically lethal. (**A**) Schematic diagram of the targeted Fdx1 gene locus and the cassette used to target the locus using a knock-out first approach. (**B**) Representative image of a PCR gel used to genotype WT and $Fdx1^{+/-}$ +/-. (**C**) The number and % of embryos and live off-springs from the intercrosses of $Fdx1^{+/-}$ +/-. (**D**) Representative images of WT, $Fdx1^{+/-}$, and $Fdx1^{-/-}$ embryos obtained at embryonic day 10.5. The white dots seen in the pictures are the result of light reflection while taking the photos. (**E**) Agarose gel image of a PCR used to genotype WT, $Fdx1^{+/-}$, and $Fdx1^{-/-}$ embryos shown in (**D**).

The online version of this article includes the following source data for figure 1:

**Source data 1.** Unedited DNA gel images for *Figure 1B and E*.

## Mice deficient in $Fdx1$ have shorter lifespan and are prone to steatohepatitis/steatosis

Although $Fdx1^{-/-}$ mice were embryonically lethal, $Fdx1^{+/-}$ +/- were viable. Thus, to study the effect of Fdx1 loss on the long-term survival as well as predisposition to tumors and other abnormalities, we generated a cohort of $Fdx1^{+/-}$ +/-. Since Fdxr catalyzes the transfer of an electron from NADPH to Fdx1, we compared the phenotypes between $Fdx1^{+/-}$ +/- and $Fdxr^{+/-}$ +/- from our previous studies

(*Zhang et al., 2022*; *Zhang et al., 2017*). To minimize the number of animals used, the data for the *WT* mice were adapted from two previous studies (*Yang et al., 2017*; *Zhang et al., 2017*), as all the mice were derived from the same C57BL/6 background and maintained in the same facility. Moreover, to avoid any influences from sex differences, all the mice cohorts were kept at close to equal ratio of male:female mice.

Previously, we found that $Fdxr^{+/-}$ +/- had a short life span compared to WT mice (*Zhang et al., 2022*; *Zhang et al., 2017*). Here, we found that the median lifespan for $Fdx1^{+/-}$ +/- (73 weeks) was significantly shorter than that for WT mice (117 weeks) and even shorter than that for $Fdxr^{+/-}$ +/- (102 weeks) (*Figure 2A* and *Figure 2—source data 3*). To examine if the $Fdx1^{+/-}$ +/- were tumor-prone, we performed histological analysis. $Fdxr^{+/-}$ (26 out of 29) mice were tumor-prone as compared to WT mice (11 out of 51) (p<0.00001 by Fisher's exact test). However, the overall incidence of spontaneous tumors was not statistically significant between $Fdx1^{+/-}$ +/- (11 out of 26) and WT mice (11 out of 51) (p=0.0673 by Fisher's exact test) (*Figure 2B*, *Figure 2—source data 3*). Nevertheless, like $Fdxr^{+/-}$ +/-, $Fdx1^{+/-}$ +/- showed significantly higher incidences of adenocarcinomas and sarcomas as compared to WT mice (*Figure 2B–D*). Out of the three Fdx1+/- mice that developed adenocarcinomas, two had lung adenocarcinomas and one had adenocarcinoma associated with the gastrointestinal tract (*Figure 2D*, *Figure 2—source data 3C*).

Previously, we showed that mice deficient in *Fdxr* were prone to liver steatosis and inflammation (*Figure 2E*, *Figure 2—source data 3A,B*; *Zhang et al., 2022*; *Zhang et al., 2017*). Since Fdxr and Fdx1 are necessary for steroidogenesis/lipid metabolism, we next examined whether $Fdx1^{+/-}$ mice are susceptible to liver abnormalities. Indeed, we found that like $Fdxr^{+/-}$ +/-, $Fdx1^{+/-}$ +/- were highly prone to liver steatosis/steatohepatitis (*Figure 2E–F*, *Figure 2—source data 3C*). As steatohepatitis often leads to liver cirrhosis and eventually liver failure (*Sanches et al., 2015*), we conclude that the high incidence of steatosis/steatohepatitis are likely responsible for the shorter lifespan observed in $Fdx1^{+/-}$ +/-. Of note, we did not observe any variations in phenotypes based on any sex differences in our studies.

## Lack of FDX1 leads to altered lipid metabolism possibly via the ABCA1-SREBP1/2 pathway

We previously showed that FDXR regulates lipid metabolism via SREBP1/2 pathways (*Zhang et al., 2022*). Given that FDX1 is a substrate of FDXR during steroidogenesis, FDX1 might be involved in lipid metabolism via the same pathway. To test this, we generated $Fdx1^{+/-}$ mouse embryo fibroblasts (+/-). As $Fdx1^{+/-}$ +/- are lethal at E10.5, we were only able to obtain $Fdx1^{+/-}$ +/-. As expected, the levels of Fdx1 protein in $Fdx1^{+/-}$ +/- were approximately half of that in WT MEFs (*Figure 3A*). To examine whether loss of FDX1 alters lipid metabolism, Nile Red (9-diethylamino-5H-benzo[a]phenoxazine-5-one) staining, which stains lipid droplets, was performed on these cells. We found intense Nile red staining in the cytoplasm of $Fdx1^{+/-}$ +/-, suggesting accumulation of lipid droplets in *Fdx1*-deficient MEFs as compared to WT MEFs (*Figure 3B*).

To confirm the role of Fdx1 in lipid metabolism in other murine cells, we used CRISPR-Cas9 to knock-out *Fdx1* in mouse FL83B hepatocytes and SCp2 mammary epithelial cells. Unfortunately, we were not able to obtain any *Fdx1*-KO clones in both cell lines, indicating a requirement of Fdx1 for cell survival (*Figure 3C and G*). Interestingly, along with the expected decrease in Fdx1 protein levels, we observed lower levels of Fdxr protein in $Fdx1^{+/-}$ +/- as compared to isogenic control cells, suggesting that Fdx1 regulates Fdxr expression (*Figure 3G and H*). Next, we performed Nile red staining to examine lipid droplet accumulation in these cells. We found that loss of Fdx1 led to increase in the accumulation of lipid droplets in the cytoplasm of FL83B and SCp2 cells (*Figure 3D and I*), consistent with the observation in MEFs that loss of Fdx1 leads to aberrant lipid metabolism.

To investigate whether Fdx1 modulates lipid metabolism via the ABCA1-SREBP1/2 pathway, we cultured FL83B cells in serum-free medium, which mimics depletion of cholesterol response. First, we examined whether Fdx1 deficiency has an effect on the level of ABCA1, a cholesterol efflux pump, which is known to be regulated upon Fdxr deficiency (*Yamauchi et al., 2015*). We found that Fdx1 deficiency led to decreased expression of ABCA1 protein (*Figure 3E and F*). Moreover, the mature nuclear form of SREBP2 protein, which is known to be increased once ABCA1 expression is decreased (*Moon et al., 2019*), was increased upon Fdx1 deficiency (*Figure 3E and F*). Furthermore, we examined if Fdx1 deficiency affects the maturation of SREBP1, which is mainly involved in the fatty acid

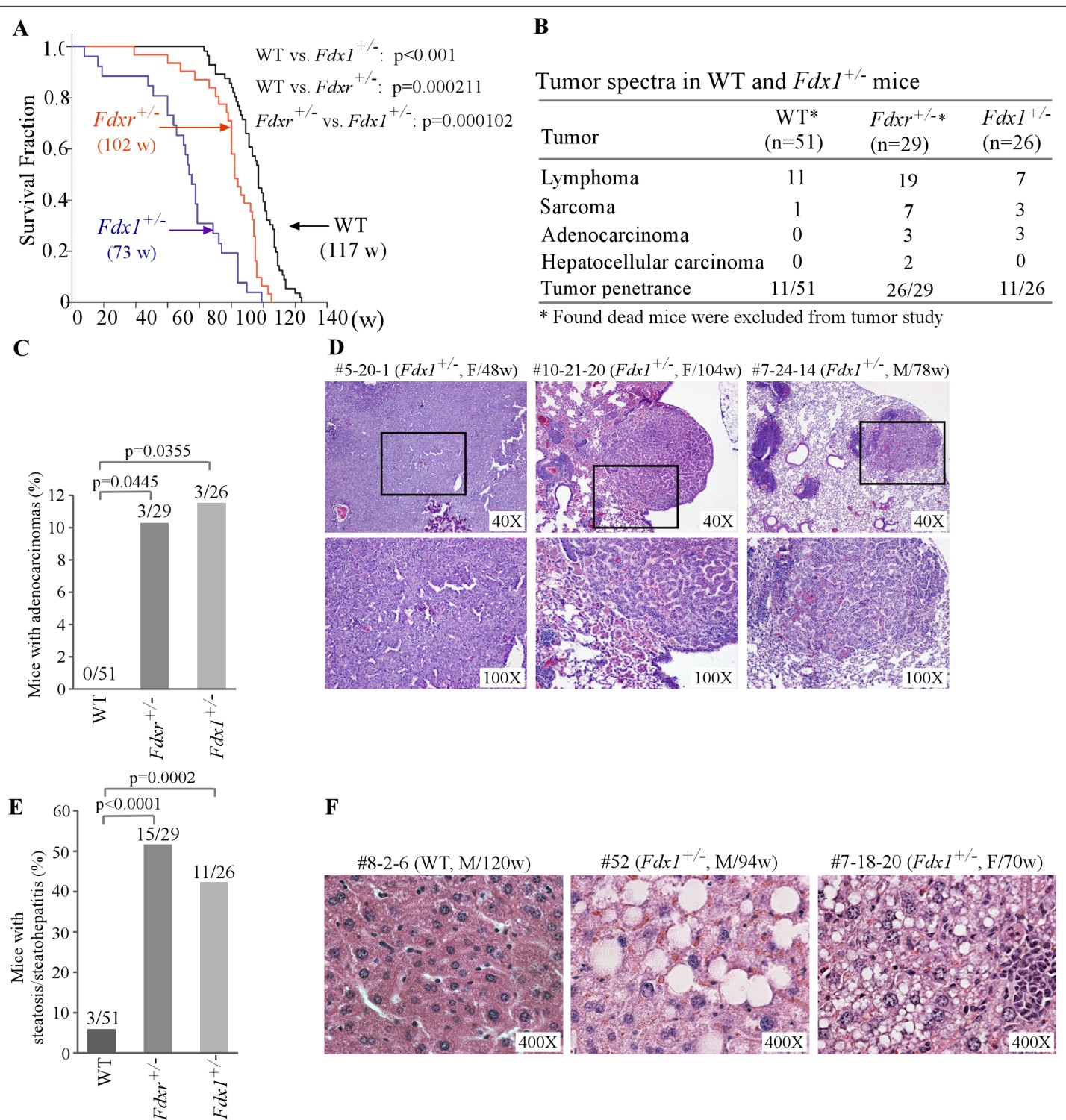

**Figure 2.** *Fdx1⁺ᐟ⁻* +/- have shorter life-span and increased incidence of adenocarcinoma and steatohepatitis. (**A**) Kaplan-Meier survival curves of WT (n=56), *Fdxr⁺ᐟ⁻* (n=31), and *Fdx1⁺ᐟ⁻* (n=26) mice. (**B**) Tumor spectra and penetrance in WT, *Fdxr⁺ᐟ⁻*, and *Fdx1⁺ᐟ⁻* +/-. (**C**) The numbers and percentages of WT, *Fdxr⁺ᐟ⁻*, and *Fdx1⁺ᐟ⁻* +/- with adenocarcinoma. (**D**) Representative images of hematoxylin and eosin (H&E)-stained adenocarcinomas from Fdx1+/-mice. The images are of gastrointestinal tract-associated adenocarcinoma in mouse #5-20-1, whereas they are of lung adenocarcinoma in mice #10-21-20 and #7-24-14. (**E**) The numbers and percentages of WT, *Fdxr⁺ᐟ⁻*, and *Fdx1⁺ᐟ⁻* +/- with liver steatosis/steatohepatitis. (**F**) Representative images of hematoxylin and eosin (H&E)-stained WT and *Fdx1⁺ᐟ⁻* +/- showing steatosis in *Fdx1*-deficient mice.

The online version of this article includes the following source data for figure 2:

*Figure 2 continued on next page*

*Figure 2 continued*

**Source data 1.** Kaplan-Meier survival analysis for various mice cohorts in *Figure 2A*.

**Source data 2.** Analysis, graphs, and statistical significance for *Figure 2C and E*.

**Source data 3.** Wild type (WT) mice (n=56) - survival time, tumor spectrum, steatosis, inflammation, and other abnormalities (A); *Fdxr+/-* mice (n=31) - survival time, tumor spectrum and other abnormalities (B); *Fdx1+/-* mice (n=26) - survival time, tumor spectrum, steatosis, inflammation, and other abnormalities (C).

and triacylglyceride (TAG) synthesis (*Shimano et al., 1997*; *Shimano et al., 1996*). We found that the mature form of SREBP1 was only slightly increased in Fdx1-deficient cells (*Figure 3E and F*). As the maturation of SREBP2 was very robust in Fdx1-deficient cells, we further tested the induction of mevalonate decarboxylase (MVD), an SREBP2 target involved in ATP-dependent decarboxylation of mevalonate pyrophosphate to form isopentenyl pyrophosphate (*Hinson et al., 1997*; *Michihara et al., 2001*). We found that MVD was increased in Fdx1-deficient cells (*Figure 3E and F*), indicating that the mature nuclear form of SREBP2 protein induced by Fdx1 deficiency is transcriptionally active.

Next, we used HCT116 human colorectal cancer cells to determine if the effect of FDX1 on lipid metabolism observed in murine cells is conserved in human cells. We again employed the CRISPR-Cas9 system to knock out the *FDX1* gene in HCT116 cells (*Zhang et al., 2017*). Surprisingly, unlike murine cells, we were able to obtain complete knock-out of *FDX1* in human cells (*Figure 4A*). Nevertheless, we found that like in murine cells, loss of FDX1 led to increased activation of the SREBP1/SREBP2 pathways in HCT116 cells (*Figure 4B–C*). Furthermore, lipid droplets were increased in the *FDX1*-KO HCT116 cells compared to isogenic control cells (*Figure 4D*). Since Nile Red is not specific to certain lipids and can recognize several intracellular neutral lipids, including TAGs (*Greenspan et al., 1985*) and sterol esters/cholesterol esters (CEs) (*Genicot et al., 2005*), in vitro enzymatic assays were performed to specifically measure the levels of TAGs and cholesterol in these cells. We found that the intracellular levels of both cholesterol and TAGs were significantly upregulated in *FDX1*-KO HCT116 cells compared to isogenic control cells (*Figure 4E and F*), confirming aberrant lipid metabolism upon loss of FDX1.

As loss of FDX1 promotes lipid droplet accumulation, we wanted to investigate if overexpression of FDX1 would decrease lipid droplet accumulation. To test this, FDX1 was inducibly expressed under the control of the doxycycline-regulated promoter in HCT116 cells. Treatment with doxycycline led to robust overexpression of FDX1 in these cells (*Figure 4—figure supplement 1A*). Next, we examined lipid droplet accumulation in these cells with or without FDX1 induction using Nile red staining. We found that overexpression of FDX1 decreased lipid droplet accumulation (*Figure 4—figure supplement 1B*). Together, we demonstrated that FDX1 is involved in lipid metabolism possibly via SREBP1/2 pathways in both murine and human cells.

## Loss of FDX1 alters the lipidome in HCT116 cells

As loss of FDX1 appeared to alter lipid metabolism, we wanted to examine which classes of lipids are altered by FDX1 deficiency. For this, untargeted lipidomic analysis was performed with isogenic control and *FDX1*-KO HCT116 cells. A total of 710 lipids were identified and quantified by LC-MS/MS, including cholesterol, TAGs, fatty acids, cardiolipins, acylcarnitines (CARs), ceramides (CERs), various phospholipids and lysophospholipids. Consistent with the data measured by the enzymatic assay (*Figure 4D*), untargeted lipidomic analysis showed that the levels of cholesterol were highly increased in FDX1-deficient cells as compared to that in isogenic control cells (*Figure 5A* and *Figure 5—figure supplement 1A*). In addition, loss of FDX1 led to a significant increase in the content of acylcarnitines (CARs) and phosphotidylcholines (PCs) (*Figure 5C–D*; *Figure 5—figure supplement 1B-C*) but a marked decrease in lysophosphotidylcholine (LPCs), lysophosphotidylethanolamine (LPEs) and ceramides (CERs) (*Figure 5E–G*; *Figure 5—figure supplement 1D-F*). On the other hand, we did not observe any significant difference in the levels of phosphotidylethanolamines (PEs), cardiolipins, sphingomyelins (SMs), fatty acids (FAs), and diacylglycerols (DAGs) (*Figure 5H–L*). Interestingly, while the level of TAGs was found to be increased by the enzymatic assay (*Figure 4E*; *Figure 5—figure supplement 1G*), the level of TAGs was found to be decreased by untargeted lipidomic analysis (*Figure 5B*; *Figure 5—figure supplement 1G*), suggesting that glycerols from other complex lipids may have been recognized by the enzymatic assay as TAGs (*Kapoor and Gupta, 2012*; *Haupt and*

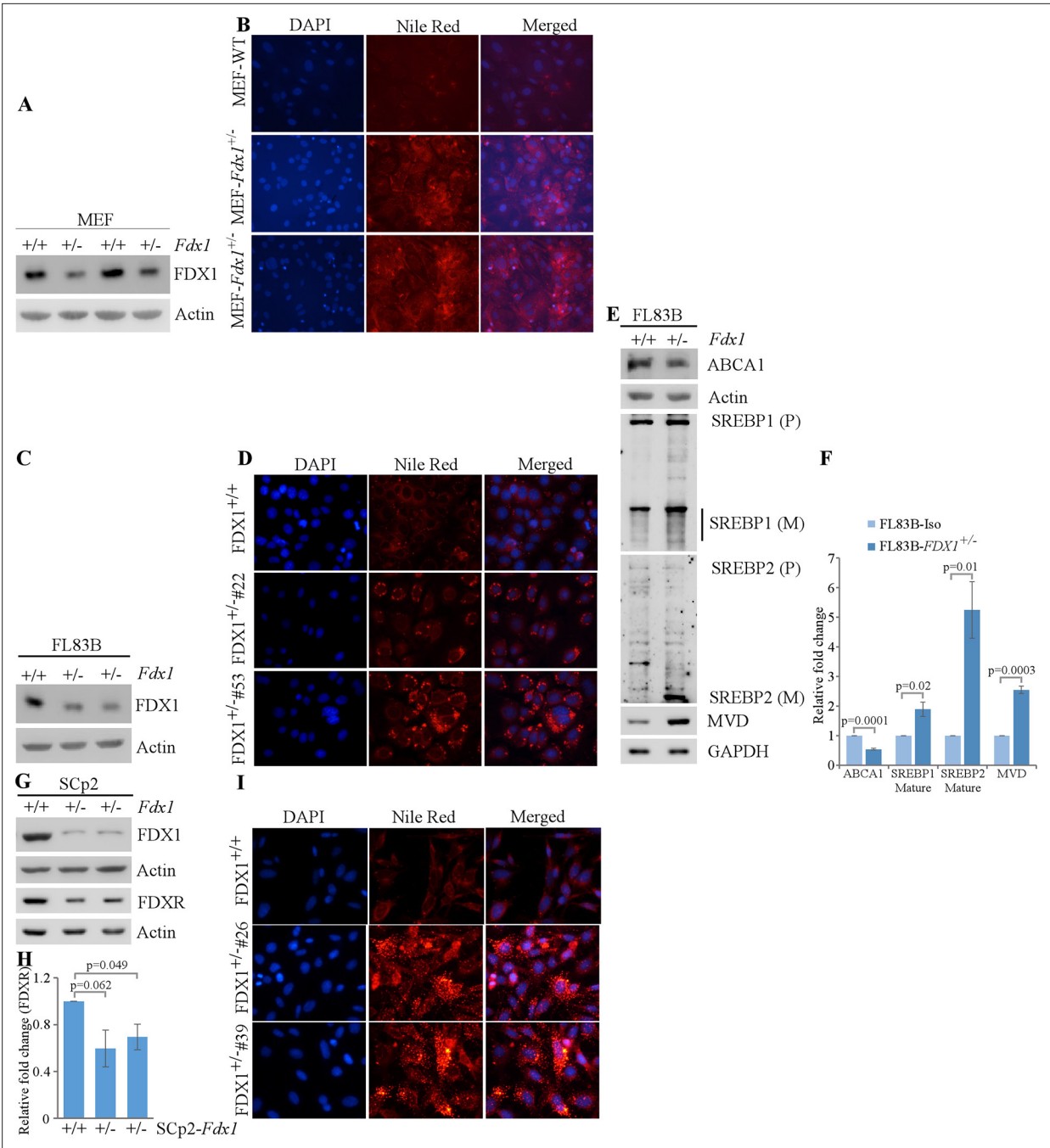

**Figure 3.** Lack of FDX1 leads to altered lipid metabolism possibly via ABCA1-SREBP1/2 pathways in murine cells. (**A**) The levels of Fdx1 and Actin were measured in two sets of WT and $Fdx1^{+/-}$+/-. (**B**) Lipid droplets were visualized with Nile Red staining (ex: 488 nm, em: 565 nm) in WT and $Fdx1^{+/-}$ +/- cultured in serum-free media for 4 hr. DAPI (ex: 358 nm, em: 461 nm) was used to stain nuclei. (**C, G**) The levels of Fdx1, Fdxr and Actin protein were measured by immunoblotting in isogenic control and two $Fdx1^{+/-}$+/-83 B cells (**D**) or SCp2 cells (**G**). (**D, I**) Isogenic control and two $Fdx1^{+/-}$+/-83B clones (clone#53 and #22) (**E**) or isogenic control and two $Fdx1^{+/-}$+/-2 clones (clone#4 and #26) were cultured in serum-free media for 4 hr followed by Nile Red staining. DAPI was used to stain nuclei. (**E**) The levels of ABCA1, SREBP1/2, MVD, and Actin were measured in isogenic control and $Fdx1^{+/-}$+/-83B cells cultured in serum-free media for 4 hr. (**F, H**) Quantification of Western blot bands from E (**F**) or G (**H**) using ImageJ. The graphs show relative fold change of the indicated proteins from at least three independent experiments. Data represent the mean ± SEM.

The online version of this article includes the following source data for figure 3:

**Source data 1.** Unedited western blot images for *Figure 3A, C, E and G*.

**Source data 2.** ImageJ analysis and statistical significance for *Figure 3F and H*.

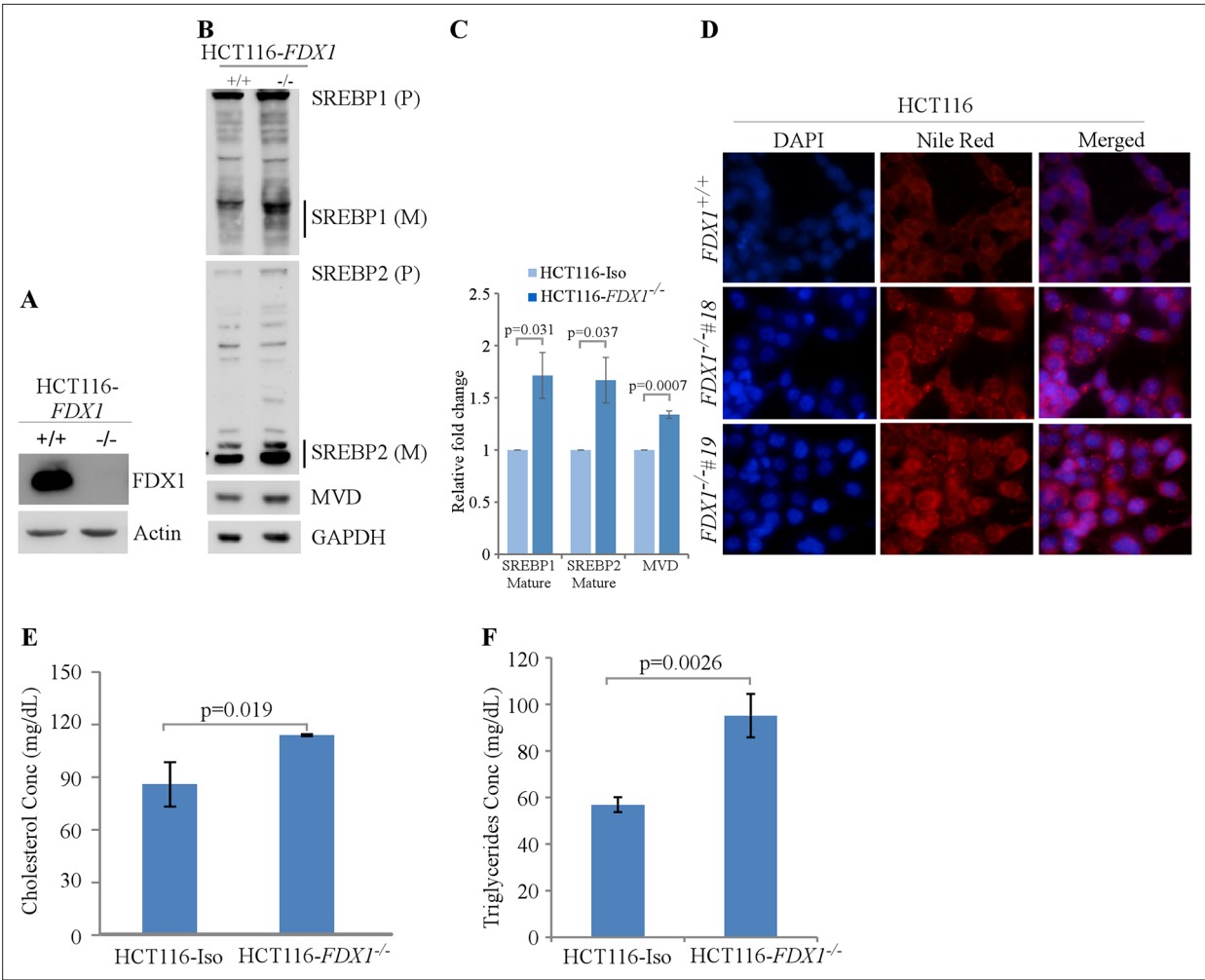

**Figure 4.** Lack of FDX1 leads to altered lipid metabolism possibly via SREBP1/2 pathways in human cells. (**A**) The levels of FDX1 and Actin protein were measured by immunoblotting in isogenic control and $FDX1^{-/-}$ HCT116 cells. (**B**) The levels of SREBP1/2, MVD, and Actin were measured in isogenic control and $FDX1^{-/-}$ HCT116 cells cultured in serum-free media for 4 hr. (**C**) Quantification of Western blot bands from (**B**) using ImageJ. The graph shows relative fold change of the indicated proteins from at least 3 independent experiments. Data represent the mean ± SEM. (**D**) Lipid droplets were visualized with Nile Red staining (ex: 488 nm, em: 565 nm) in isogenic control and two $FDX1^{-/-}$ HCT116 clones (clone#18 and #19) cultured in serum-free media for 4 hr. DAPI (ex: 358 nm, em: 461 nm) was used to stain nuclei. (**E**) Quantitative measurement of intracellular cholesterol esters. Isogenic control and $FDX1^{-/-}$ HCT116 cells were cultured in a 96-well plate. After 4 hr of fasting, the level of total cholesterol was measured with Cholesterol/Cholesterol Ester-GloTM assay kit according to manufacturer's instruction. Data represent the mean ± SD. (**F**) Quantitative measurement of intracellular triglycerides. Isogenic control and $FDX1^{-/-}$ HCT116 cells were cultured in a 96-well plate. After 4 hr of fasting, the level of total triglycerides was measured with Triglyceride-GloTM assay kit according to manufacturer's instruction. Data represent the mean ± SD.

The online version of this article includes the following source data and figure supplement(s) for figure 4:

**Source data 1.** Unedited western blot images for *Figure 4A and B*.

**Source data 2.** ImageJ analysis and statistical significance for *Figure 4C*.

**Source data 3.** Analysis, graphs, and statistical significance for *Figure 4E and F*.

**Figure supplement 1.** Over-expression of FDX1 leads to decrease in cytoplasmic lipid droplets in human cells.

**Figure supplement 1—source data 1.** Unedited western blot images for *Figure 1A*.

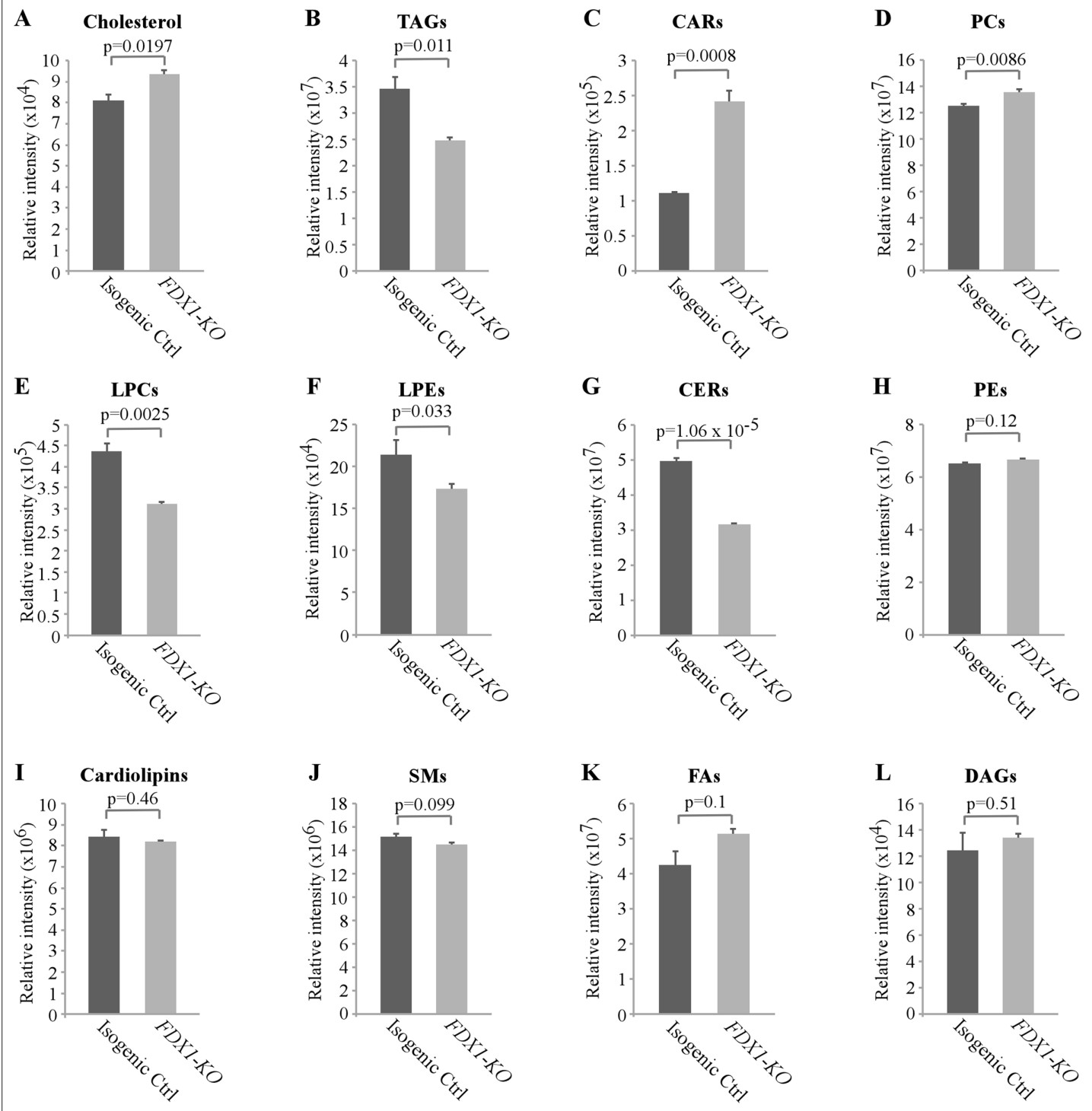

**Figure 5.** Lack of FDX1 leads to alterations in the lipid profile. Isogenic control and *FDX1*-KO HCT116 cells were used for lipidomic analysis by LC-MS/MS. The relative abundance of a lipid was calculated as mean ± SEM and statistical significance was determined using Student's t-test. (**A**) Cholesterol; (**B**) Triacylglycerides (TAGs); (**C**) Acylcarnitines (CARs); (**D**) Phosphatidylcholines (PCs); (**E**) Lysophosphatidylcholines (LPCs); (**F**) Lysophosphatidylethanolamines (LPEs); (**G**) Ceramides (CERs); (**H**) Phosphatidylethanolamines (PEs); (**I**) Cardiolipins; (**J**) Sphingomyeline (SMs); (**K**) Fatty acids (FAs); (**L**) Diacylglycerols (DAGs).

The online version of this article includes the following source data and figure supplement(s) for figure 5:

**Source data 1.** Relative lipid classes abundance, analysis, graphs and statistical significance for *Figure 5A–L*.

**Figure supplement 1.** Lack of FDX1 leads to alterations in the lipid profile.

*Valley, 2020*). Together, these data indicate that *FDX1*-KO leads to significant alterations in lipid profile due to aberrant lipid metabolism.

## Discussion

FDX1/2 transduce an electron from FDXR to various effector protein complexes. FDX1 is suggested to be involved in sterol hydroxylation and cholesterol side chain cleavage (*Schulz et al., 2023*). Recent studies indicate that FDX1 plays a critical role in the regulation of oxidative phosphorylation via lipoylation of key TCA cycle enzymes (*Schulz et al., 2023*; *Tsvetkov et al., 2022*). Although FDX1 has been extensively characterized biochemically, its role in physiology and lipid metabolism has not been explored. In this study, we generated mice deficient in *Fdx1* and showed that homozygous *Fdx1* deletion led to embryonic lethality at E10.5. We also showed that *Fdx1*$^{+/-}$ +/- had a short life span and developed steatohepatitis, which is similar to the phenotypes exhibited by *Fdxr*$^{+/-}$ +/-. Moreover, Fdx1 deficiency led to lipid droplet accumulation in both murine and human cells, possibly via the ABCA1-SREBP1/2 pathways. Consistently, untargeted lipidomic analysis showed that FDX1 deficiency led to alterations in several classes of lipids, suggesting the role of FDX1 in lipid metabolism.

The embryonic lethality by complete loss of *Fdx1* is similar to what we previously showed by complete loss of *Fdxr*, the latter of which was attributed in part to iron overload (*Zhang et al., 2017*). However, as FDX1 is not a major player in Fe-S cluster synthesis (*Schulz et al., 2023*), the lethality could be due to lack of progesterone production by placenta, which is required for the maintenance of pregnancy (*Miller and Auchus, 2011*). Thus, *Fdx1*$^{-/-}$ embryos would not be able to survive due to a miscarriage. However, since mice with complete deletion of Cyp11a1 (Cytochrome p450scc), which is essential for progesterone production, can survive till birth (*Hu et al., 2002*), we postulate that other pathways may be responsible for the lethality of *Fdx1*$^{-/-}$ embryos. Additionally, since *Fdx1*$^{-/-}$ embryos developed abnormally, we postulate that FDX1 may be required for proper embryo growth, consistent with a recent observation that loss of FDX1 would lead to defects in lipoylation of key TCA cycle enzymes and subsequent loss of oxidative phosphorylation (*Schulz et al., 2023*; *Tsvetkov et al., 2022*).

We observed that *Fdx1*$^{+/-}$ +/- were not prone to spontaneous tumors, which is different from tumor-prone *Fdxr*$^{+/-}$ +/- (*Zhang et al., 2022*; *Zhang et al., 2017*). Nevertheless, *Fdx1*$^{+/-}$ +/- exhibited a higher incidence of adenocarcinomas and sarcomas compared to WT mice (*Figure 2*). One possible explanation is that loss of *Fdxr* leads to decreased expression of p53 tumor suppressor whereas loss of Fdx1 alone does not have a significant effect on p53 expression (*Zhang et al., 2017*). We also observed that murine cells deficient in both alleles of the *Fdx1* gene are not viable, which is consistent with the observation that *Fdx1*-KO embryos die between 10.5 and 13.5 days post-fertilization (*Figure 1*). However, both our own and others' published studies showed that *FDX1*-KO human cells are viable (*Schulz et al., 2023*; *Tsvetkov et al., 2022*; *Zhang et al., 2017*). One possible explanation is that human FDX2, but not murine Fdx2, may be able to compensate for loss of FDX1 for cell survival, which is worth further investigation.

Murine and human cells deficient in FDX1 showed increased lipid accumulation possibly via activation of the ABCA1-SREBP1/2 pathways, which was again consistent with the loss of FDXR (*Zhang et al., 2022*). In order to identify specific classes of lipids altered upon loss of FDX1, lipidomic analysis was performed and showed that the levels of cholesterol, triacylglycerols, acylcarnities, phospholipids, and lysophospholipids were significantly altered by the loss of FDX1. Thus, altered lipid metabolism is likely responsible for steatohepatitis observed in Fdx1-deficient mice. Steatohepatis is a progressive disease that initiates with the accumulation of excessive fat in the liver (steatosis; *Shao et al., 2020*; *Vanni et al., 2010*). The fat accumulation would induce liver inflammation, leading to steatohepatitis, referred to as non-alcoholic steatohepatitis (NASH). These data clearly indicate that FDX1 is involved in cellular and physiological lipid homeostasis and that loss of FDX1 leads to development of pathological impairment in mice. Although previous studies showed that FDX1 is involved in cholesterol metabolism, the observations in this study raise a critical question about how FDX1 regulates the metabolism of the various classes of lipids. Thus, further studies are warranted to address this issue.

Aberrant lipid metabolism has emerged as a key player in the intricate landscape of oncogenesis, contributing to the development and progression of various cancers (*Santos and Schulze, 2012*). Aberrant regulation of lipid metabolism pathways, such as altered fatty acid synthesis, increased cholesterol metabolism, and disrupted lipid signaling, have been implicated in tumorigenesis (*Huang*

*et al., 2020*). Additionally, aberrant lipid metabolism can influence signaling pathways crucial for cell proliferation, survival, and migration (*Broadfield et al., 2021*). Although, the overall tumor penetrance in $Fdx1^{+/-}$ +/-not significantly altered as compared to WT mice, we indeed observed increased incidences of adenocarcinomas in these mice, which would signal the role of Fdx1-mediated lipid homeostasis in regulation of oncogenesis and warrants further studies.

In conclusion, we showed that FDX1 is essential for mammalian embryonic development and is also critical for the maintenance of lipid homeostasis at cellular and physiological levels.

## Materials and methods

### *Fdx1* mutant mouse model

The use of animals and the study protocols used in this article were approved by the University of California at Davis Institutional Animal Care and Use Committee under the protocol # 23011. $Fdx1^{+/-}$ +/− were generated by the Mouse Biology Program at the University of California, Davis (Davis, CA, USA). The $Fdx1^{+/-}$ +/- used in this study as well as the WT and $Fdxr^{+/-}$ mice used in the previous studies were all derived from and maintained in C57BL/6 background. The primers used to genotype the *Fdx1*-WT allele are forward primer, 5'- GGT GTA GTG TGG TGG TCA AGT ATG TG-3', and reverse primer, 5'- CAC ACC GAG GAC ATA CTC TCT CAC-3'; and for *Fdx1*-KO allele are forward primer, 5'- GGT GTA GTG TGG TGG TCA AGT ATG TG-3', and reverse primer, 5'- CTC CTA CAT AGT TGG CAG TGT TTG GG-3'.

### MEF isolation

MEFs were isolated from 12.5 to 13.5 days post-coitum (pc) mouse embryos, as described previously (*Mohibi et al., 2023*; *Mohibi et al., 2021*). To generate WT and $Fdx1^{+/-}$ +/-, $Fdx1^{+/-}$ +/- were interbred. The MEFs were cultured in DMEM supplemented with 10% FBS (Hyclone Laboratories, Erie, PA, USA), 55 μM β-mercaptoethanol and 1 x non-essential amino acids solution (Cellgro, Manassas, VA, USA).

### Cell culture

SCp2 mouse mammary epithelial cells were kindly provided by Pierre-Yves Desprez at the California Pacific Medical Center Research Institute and cultured as previously described (*Desprez et al., 1998*). Mouse normal liver FL83B cells and the human cancer cell line HCT116 were acquired from ATCC between 2007 and 2018 and utilized at passages below 20 or within 2 months of thawing. Authentication and Mycoplasma testing were not conducted since ATCC had already authenticated and tested these cell lines. This decision was particularly based on the low passages at which the cell lines were employed. FL83B and HCT116 cells were cultured in Dulbecco's modified Eagle's medium (DMEM) (Invitrogen) supplemented with 10% fetal bovine serum (FBS) (Hyclone, Logan, UT).

### Plasmid construction and cell line generation

To generate *Fdx1*-KO murine cells, two single-guide RNA (sgRNA) expression vectors pSpCas9(BB)–2A-Puro-sgFdx1-1 and pSpCas9(BB)–2A-Puro-sgFdx1-2 were used to remove initiation codon in exon 1 and create frame shift deletions. The generation of sgRNA expression vector was performed as described previously (*Ran et al., 2013*). The oligonucleotides for sgFdx1-1 are sense, 5'- CAC CGG GAC CCG GAA CCT TCC GAC –3', and antisense, 5'- AAA CGT CGG AAG GTT CCG GGT CCC –3'; for sgFdx1-2 are sense, 5'- CAC CGT CCG CGG CCT TGA CCG CTG T –3', and antisense, 5'- AAA CAC AGC GGT CAA GGC CGC GGA C –3'.

Generation of $Fdx1^{+/-}$ +/-2 and FL83B cell lines was achieved by transfecting the above guides in these cells using JetPRIME transfection reagent. The cells were selected with puromycin, and individual clones picked, genotyped, sequenced, and confirmed by western blot analysis. The primers used for genotyping mouse *Fdx1* were: forward primer, 5'- TTA TAG GAC ACG CAG GGC –3', and reverse primer, 5'- ACG GAG AGA TGG CAG TTC –3'. The generation of *FDX1*-KO HCT116 has been described previously (*Zhang et al., 2017*).

### Western blot analysis

Western blotting was performed as previously described (*Mohibi et al., 2020*). Briefly, cell lysates were collected as indicated, resolved on 8–11% SDS-polyacrylamide gels and transferred to nitrocellulose

membrane. After the transfer, the membranes were blocked with PBST containing 2.5% milk at room temperature (RT) for 1hr, followed by overnight 4 °C incubation with primary antibodies prepared in 2.5% milk containing PBST. The following day, after 3 X washings with PBST, the blots were incubated at room temperature for 1 hr in secondary antibody prepared in PBST containing 2.5% milk. After 3 X washings with PBST, the blots were soaked in enhanced chemiluminescence reagents (Thermo Fisher Scientific), and then visualized with the BioSpectrum 810 Imaging System (UVP LLC, Upland, CA). The blots from at least three independent experiments were quantified by ImageJ using standard protocol (ImageJ.pdf yorku.ca). Antibody against human/mouse ABCA1 (#PA1-16789; RRID:AB_2288917) was purchased from Invitrogen Life Technologies (Carlsbad, CA) and Cell signaling technology (#96292, Danvers, MA). Antibodies against SREBP1 (#ab3259; RRID:AB_303650), SREBP2 (#ab30682; RRID:AB_779079; used to detect human SREBP2) and FDX1 (#ab108257) were purchased from Abcam (Cambridge, MA). Antibody against SREBP2 (#MAB7119, to detect mouse SREBP2) was purchased from R&D systems (Minneapolis, MN). Antibody against FDXR (#sc-374436) was purchased from Santa Cruz Biotechnology (Dallas, TX). HRP-conjugated secondary antibodies against rabbit and mouse IgG were purchased from BioRad (Hercules, CA).

### RNA isolation and RT-PCR analysis

Total RNA was harvested using TRIzol reagent (Invitrogen) and isolated according to the manufacturer's instructions. RevertAid First Strand cDNA Synthesis kit (Thermo Fisher Scientific) was used to synthesize cDNA from 2 µg total RNA as per manufacturer's protocol. The primers used to amplify mouse AFP were forward primer, 5'- AGG AGG AGT GCT TCC AGA CA –3', and reverse primer, 5'-TGC GTG AAT TAT GCA GAA GC –3'. The primers used to amplify mouse actin were forward primer, 5'-TCC ATC ATG AAG TGT GAC GT-3', and reverse primer, 5'-TGA TCC ACA TCT GCT GGA AG-3'.

### Nile Red staining

Nile Red (#ab219403) was purchased from Abcam (Cambridge, MA). A 10 µM stock solution of Nile red was prepared by dissolving it in acetone. Nile red was diluted to 2 µg/mL in complete medium and added to the cells for 30 min at room temperature. Subsequently, the cells were washed with PBS and fixed in 4% paraformaldehyde (Sigma Aldrich, St Louise, MO) for 20 min at room temperature. The cells were then counterstained with DAPI and images were obtained using with confocal microscope.

### Measurement of cellular cholesterol and triglycerides

Total cholesterol and triglycerides from cells were determined by using Cholesterol/Cholesterol Ester-Glow assay kit (Promega, Madison, WI) and Triglyceride-Glow assay kit (Promega, Madison, WI), respectively. Briefly, cells were plated in 96-well plate. After 4 hr of fasting in serum-free media, the levels of cholesterol and triglycerides were measured according to manufacturer's protocol.

### Histological analysis

Wild-type, *Fdxr*$^{+/-}$, and *Fdx1*$^{+/-}$ mouse tissues were fixed in 10% (w/v) neutral buffered formalin, processed, and embedded in paraffin blocks. Embedded tissues were sectioned (6 µm) and stained with H&E.

### Untargeted lipidomic analysis using LC-MS/MS

Lipids were extracted and analyzed by reversed-phase liquid chromatography tandem mass spectrometry (RPLC-MS/MS) at West Coast Metabolomics Center with published methods (*Ding et al., 2021*). Thirty million isogenic control or *FDX1*-KO HCT116 cells were harvested and analyzed for lipidomics as previously described (*Rabow et al., 2022*).

### Statistical analysis

The data were presented as Mean ± SEM or Mean ± SD as indicated. Statistical significance was determined by two-tailed Student's *t* test. Values of $p < 0.05$ were considered significant. For Kaplan-Meyer survival analysis, log-rank test was performed. Fisher's exact test was used for comparison of tumors, adenocarcinomas, and liver steatosis/steatohepatitis from different mice cohorts.

## Acknowledgements

This work is supported in part by NIH grant CA224433.

## Additional information

### Funding

| Funder | Grant reference number | Author |
|---|---|---|
| National Institutes of Health | CA224433 | Xinbin Chen |

The funders had no role in study design, data collection and interpretation, or the decision to submit the work for publication.

### Author contributions

Shakur Mohibi, Data curation, Investigation, Writing - original draft, Writing – review and editing; Yanhong Zhang, Conceptualization, Data curation, Investigation, Methodology; Vivian Perng, Data curation; Mingyi Chen, Formal analysis; Jin Zhang, Resources, Data curation, Supervision, Methodology, Writing – review and editing; Xinbin Chen, Conceptualization, Resources, Supervision, Funding acquisition, Methodology, Writing – review and editing

### Author ORCIDs

Shakur Mohibi ⓘ http://orcid.org/0000-0001-5539-0841
Mingyi Chen ⓘ http://orcid.org/0000-0001-6754-0480
Jin Zhang ⓘ http://orcid.org/0000-0002-6835-920X
Xinbin Chen ⓘ http://orcid.org/0000-0002-4582-6506

### Ethics

All animal procedures were approved by UC Davis IACUC in adherence to the NIH "Guide for the Care and Use of Laboratory Animals". under the protocol # 23011.

### Decision letter and Author response

Decision letter https://doi.org/10.7554/eLife.91656.sa1
Author response https://doi.org/10.7554/eLife.91656.sa2

## Additional files

### Supplementary files

• MDAR checklist

### Data availability

The authors confirm that the data supporting the findings of this study are available within the article, its supplementary materials and source data files.

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
