## [Editor Report]

The findings provided by Mohibi et al. are important to the field of lipid metabolism and cancer and provide insight for an in vivo role of FDX1. The evidence is solid, utilizing multiple modalities and both in vitro and in vivo lines of investigation.

---

## [Decision Letter]

**Decision letter after peer review:**

Thank you for submitting your article "Ferredoxin 1 is essential for embryonic development and lipid homeostasis" for consideration by *eLife*. Your article has been reviewed by 2 peer reviewers, one of whom is a member of our Board of Reviewing Editors, and the evaluation has been overseen by David James as the Senior Editor.

Essential revisions:

1) Quantification of western blots and statistical analysis is needed.

*Reviewer #2 (Recommendations for the authors):*

– Please provide a rationale to examine AFP mRNA expression. It would also be great if the authors can perform RT-qPCR to quantify and WB detection of AFP protein expression.

– Figure 3, the statement of "Fdx1+/- mice is not alive" is not correct. (Figure 2) Theoretically, you can generate Fdx-/- MEF as you do have Fdx1-/- embryo at D10.5. However, since you already have a strong phenotype for the Fdx1+/- MEF, I would be fine to not perform additional experiments using Fgx1-/- MEF cells (although the Fdx1-/- MEF would give an even stronger phenotype in lipid metabolism alterations).

– Figure 1D, the white dots are misleading in the images. Do you have images without the dots? If not, please explain the dots (light reflection?).

---

## [Author Response]

Essential revisions:1) Quantification of western blots and statistical analysis is needed.

We thank the editor and the reviewers for their recommendation to quantify the western blots and provide statistical analysis. We have now provided quantifications of the western blots from old Figure 3F, 3G and 4B and also changed some of the blots with clear and convincing data (Please see revised Figures 3E, 3F, 3G, 3H, 4B and 4C). We hope that the reviewers and editors could find these data more convincing.

Reviewer #2 (Recommendations for the authors):– Please provide a rationale to examine AFP mRNA expression. It would also be great if the authors can perform RT-qPCR to quantify and WB detection of AFP protein expression.

We thank the reviewer for their suggestion. As mentioned in the manuscript, increased expression of α-fetoprotein is associated with inflammation and cancers, including liver, kidney and lymphoma (Galle et al., 2019). The levels of α-fetoprotein in blood has been used as important screening, prognostic as well as recurrence marker for hepatocellular carcinoma (Galle et al., 2019). Additionally, the mRNA levels of α-fetoprotein in blood have been shown to correlate with HCC patient outcome and recurrence (Jeng et al., 2004; Matsumura et al., 1999; Morimoto et al., 2005). As *Fdx1^+/-^* mice developed liver abnormalities, we tested the AFP mRNA levels in *Fdx1^+/-^* MEFs. However, as the levels of AFP in MEFs may or may not correlate with AFP mRNA levels in blood of *Fdx1^+/-^* mice, we have decided to remove these data from the revised manuscript.

– Figure 3, the statement of "Fdx1+/- mice is not alive" is not correct. (Figure 2) Theoretically, you can generate Fdx-/- MEF as you do have Fdx1-/- embryo at D10.5. However, since you already have a strong phenotype for the Fdx1+/- MEF, I would be fine to not perform additional experiments using Fgx1-/- MEF cells (although the Fdx1-/- MEF would give an even stronger phenotype in lipid metabolism alterations).

We thank the reviewer for their suggestion and agree that it might be possible to isolate *Fdx1^-/-^* MEFs at E10.5. However, as we were only able to obtain and study *Fdx1^+/-^* mice for various phenotypes, we believe that MEFs with the same genotype (*Fdx1^+/-^*) were more suitable to study the mechanism of action leading to those phenotypes.

– Figure 1D, the white dots are misleading in the images. Do you have images without the dots? If not, please explain the dots (light reflection?).

We understand reviewer’s concerns about the white dots and thank them for pointing this out. Those white dots are indeed light reflections that appeared while taking the pictures. Unfortunately, as we obtained only one *Fdx1^-/-^* pup at E10.5, we don’t have another image to replace. Nonetheless, as suggested by the reviewer, we have now included an explanation for this in the figure legend.

References:

Galle, P.R., Foerster, F., Kudo, M., Chan, S.L., Llovet, J.M., Qin, S., Schelman, W.R., Chintharlapalli, S., Abada, P.B., Sherman, M., et al. (2019). Biology and significance of α-fetoprotein in hepatocellular carcinoma. Liver Int. *39*, 2214–2229.

Jeng, K.S., Sheen, I.S., and Tsai, Y.C. (2004). Circulating messenger RNA of α-fetoprotein: A possible risk factor of recurrence after resection of hepatocellular carcinoma. Arch. Surg. *139*, 1055–1060.

Matsumura, M., Shiratori, Y., Niwa, Y., Tanaka, T., Ogura, K., Okudaira, T., Imamura, M., Okano, K.I., Shiina, S., and Omata, M. (1999). Presence of α-fetoprotein mRNA in blood correlates with outcome in patients with hepatocellular carcinoma. J. Hepatol. *31*, 332–339.

Morimoto, O., Nagano, H., Miyamoto, A., Fujiwara, Y., Kondo, M., Yamamoto, T., Ota, H., Nakamura, M., Wada, H., Damdinsuren, B., et al. (2005). Association between recurrence of hepatocellular carcinoma and α-fetoprotein messenger RNA levels in peripheral blood. Surg. Today *35*, 1033–1041.